# A comprehensive 22-year global GNSS climate data record from 5085 stations

Xiaoming Wang<sup>1,2</sup>, Haobo Li<sup>3,\*</sup>, Suelynn Choy<sup>3</sup>, Qiuying Huang<sup>1</sup>, Wenhui Cai<sup>3</sup>, Anthony Rea<sup>3</sup>, Hongxin Zhang<sup>1</sup>, Luis Elneser<sup>3</sup>, and Yuriy Kuleshov<sup>4</sup>

<sup>1</sup>Aerospace Information Research Institute, Chinese Academy of Sciences, Beijing 100094, China
 <sup>2</sup>University of Chinese Academy of Sciences, Beijing 100049, China
 <sup>3</sup>RMIT University, Melbourne, VIC 3001, Australia
 <sup>4</sup>Bureau of Meteorology, Melbourne, VIC 3008, Australia

Correspondence to: Haobo Li (haobo.li@rmit.edu.au)

- Abstract. This work presents a comprehensive global GNSS climate data record derived from 5085 stations, spanning a 22year time period 2000–2021. Generated through the GPAC-Repro campaign, the dataset utilises state-of-the-art processing methodologies and precise products from the International GNSS Service (IGS) Repro-3 initiative. The dataset includes highquality hourly estimates of Zenith Total Delay (ZTD) and Precipitable Water Vapour (PWV), offering improved accuracy and spatiotemporal coverage. A rigorous data screening and quality assessment framework was implemented, including formal
- error detection, offset identification, and extensive cross-validation with ERA5 reanalysis dataset, radiosonde profiles, and Very Long Baseline Interferometry (VLBI) measurements. Collectively, these efforts ensured the consistency, accuracy, and homogeneity of the dataset. In addition, diurnal, monthly, and annual variations in ZTD and PWV have been analysed to evaluate and demonstrate its feasibility for monitoring climate variability, atmospheric circulation, and weather extremes. The insights provided by the dataset address critical data gaps in global climate observing systems and provide a robust foundation
- for advancing climate research and applications. Representing a significant milestone in GNSS climatology, this dataset serves as a vital resource for the scientific community, supporting improved understanding of atmospheric processes and more effective responses to climate-related challenges.

**Keywords**: Global Navigation Satellite Systems (GNSS), precipitable water vapour (PWV), zenith total delay (ZTD), satellite Earth observation, GNSS climatology, atmospheric monitoring

# 25 1 Introduction

We are currently experiencing an alarming rise in global temperatures and an accelerated progression of climate change, manifesting in increasingly severe and frequent weather and climate extremes across the planet (Seneviratne et al., 2021). The repercussions of these events are profound, causing significant adverse socioeconomic consequences and posing substantial challenges to the sustainable development of human society. It is estimated that around 3.5 billion people are highly vulnerable

- to climate change, with over 1.5 billion already affected by weather and climate extremes (Asian Disaster Reduction Centre, 2015). Additionally, economic losses attributed to these extreme events now exceed \$1.3 trillion annually (Calvin et al., 2023). Overall, the growing body of evidence on observed impacts and the escalating trend of disasters highlight a rapidly diminishing window of opportunity to enable progress towards constructing climate-resilient communities. Despite global efforts spanning several decades, considerable data gaps remain, particularly in the existing climate observing networks. It is therefore
- important to generate long-term, homogeneous datasets for Essential Climate Variables (ECVs) to deepen our comprehension of the intrinsic nature of weather and climate extremes and enhance comprehensive climate services for the benefit of current and future generations (Bojinski et al., 2014).

Among the various atmospheric parameters, water vapour, recognised as an ECV, plays a significant role in studying global climate change and atmospheric variability (Dessler et al., 2008; Solomon et al., 2010; Labbouz et al., 2015; Ye et al., 2015).

- Substantial evidence also demonstrates that the dynamic movement of water vapour directly drives meteorological fluctuations. Consequently, access to accurate and timely water vapour data is crucial for enhancing the robustness of climate models and improving assessment of climate risks. Since the 1940s, radiosondes have been deployed to monitor atmospheric conditions and derive accurate water vapour measurements (Brettle and Galvin, 2003; Durre et al., 2006). However, these sounding balloons, typically launched twice daily from sparsely distributed stations around the globe, offer observations with limited
- spatiotemporal resolution (Li et al., 2003; Benjamin et al., 2004; Liu et al., 2013). In addition to radiosondes, water vapour radiometers and satellite-based instruments have been adopted to measure atmospheric water vapour content (England et al., 1993; Buehler et al., 2008). While widely adopted, these technologies face certain challenges, including high operational costs, limited temporal and vertical resolution, low precision, and susceptibility to weather conditions (Elliott, 1995; Gui et al., 2017). Given the limitations, there is a strong rationale for adopting an emerging technology, i.e., Global Navigation Satellite Systems
- (GNSS), for additional remote sensing of atmospheric water vapour. Initially designed for positioning, navigation, and timing, GNSS technology, like the Global Positioning System (GPS) has broadened its applications to include atmospheric monitoring since the 1990s (Elgered et al., 1991; Bevis et al., 1992; Duan et al., 1996). In ground-based GNSS atmospheric monitoring, GNSS receivers function as atmospheric sensors by tracking changes in signals as they traverse the atmosphere. Variations in water vapour, pressure, and temperature in the troposphere significantly affect the speed and trajectory of these GNSS signals,
- causing propagation delays. By measuring and analysing these signal delays from satellites to GNSS receivers, atmospheric parameters, like zenith total delay (ZTD) and precipitable water vapour (PWV), can be estimated (Rocken et al., 1993, 1995; Nilsson and Elgered, 2008; Wang et al., 2017). With used together with conventional techniques, the distinct advantages of GNSS atmospheric data, including high-accuracy, high spatiotemporal resolution, long-term stability, broad-coverage and all-weather capability, unequivocally enhance the potential for advancing weather and climate research and improving response
- to climate risks (Gradinarsky et al., 2002; Jin et al., 2007; Choy et al., 2011; Jones et al., 2020; Li et al., 2020, 2023a, 2023b). In recent years, the innovative utilisation of GNSS-derived ZTD and PWV estimates has spurred the development of various statistical, numerical, and artificial intelligence (AI)-empowered approaches for nowcasting and very short-range forecasting of weather extremes, such as heavy precipitation and tropical cyclones (Zhao et al., 2018, 2022; Benevides et al., 2019; Rohm et al., 2019; Manandhar et al., 2019; Zhang et al., 2022; Li et al., 2022b, c). Beyond these meteorological applications, GNSS
- atmospheric parameters have also significantly enriched climate studies (Hagemann et al., 2003; Bock et al., 2007; Zhao et al., 2020; Ma et al., 2021; Li et al., 2022a, d). Notably, Foster et al. (2000) demonstrated that PWV effectively captured the water vapour variability induced by the 1997–1998 El Niño event. Gradinarsky et al. (2002) reported a long-term linear increase in PWV of 0.1–0.2 mm/year across Scandinavia from 1993 to 2000. Nyeki et al. (2005) highlighted that PWV could track all-weather water vapour trends, unlike precision filter radiometers, which are limited to clear-sky conditions. Further studies of
- trends in PWV series were conducted in Finland and Sweden (Nilsson and Elgered, 2008) from 1996 to 2006, in Switzerland (Morland et al., 2009) from 1996 to 2007, and in South Korea (Sohn and Cho, 2010) from 2000 to 2009. Additionally, Wang et al. (2018) applied singular spectrum analysis (Wang et al., 2016a, b) to extract nonlinear trends in PWV series, demonstrating its potential for depicting the evolution of droughts and floods. Several other studies have also explored seasonal variations in GNSS atmospheric parameters, their responses to climate change, and their feasibility in monitoring climate extremes (Jin et al., 2016).
- al., 2007; Jin and Luo, 2009; Wang and Zhang, 2009; Ning et al., 2013; Jiang et al., 2017; Li et al., 2024). Collectively, these studies underscore the key role of GNSS atmospheric parameters in advancing weather and climate research. However, despite the advances, the potential of GNSS atmospheric monitoring remains largely unutilised in the climate community, primarily due to the lack of robust long-term GNSS climate datasets and comprehensive analysis. Often, the dataset utilised in the aforementioned studies span only around 10 years, which is insufficient for uncovering the climate change signals embedded
- in these parameters. Therefore, given the continuous enhancement of multi-constellation, multi-frequency GNSS capabilities, the availability of new data streams, and the extensive accumulation of GNSS data since the 1990s, this juncture presents a prime opportunity to generate a long-term, homogeneous GNSS climate dataset, thus fully harnessing the capabilities of GNSS

atmospheric monitoring for climate applications.

- Numerous international academic organizations and many governmental stakeholders have embarked on initiatives to generate accurate GNSS atmospheric parameters, aiming to advance atmospheric and climate studies. For example, the Troposphere Working Group (TWG) of the International GNSS Service (IGS) exemplifies such efforts by producing the "final" tropospheric estimates. These parameters are processed by the United States Naval Observatory (USNO) utilising the "final" satellite, orbit, and Earth Orientation Parameters (EOP) combination products, typically made available around three weeks after observation (Byram et al., 2011). However, the determined ZTD time series may still exhibit inhomogeneities due to updates in reference
- frames and models, variations in mapping function implementations, adjustments in elevation cut-off angles, and modifications in processing strategies. For climate-related research, maintaining the homogeneity of ZTD and PWV time series is essential, as reliable climate change monitoring relies on the utilisation of robust and consistent datasets (Vey et al., 2009; Van Malderen et al., 2014; Ning et al., 2016). Therefore, to address this, it is important to reprocess long-term historical GNSS data using consistent processing strategies, including uniform mapping functions, elevation cut-off angles, and models, like phase centre
- variation. In response, the IGS analysis centres have undertaken two significant reprocessing campaigns, utilising the most recent models, updated processing strategies, and the latest satellite orbits, clock corrections, and EOP estimates. The second IGS reprocessing campaign (known as "Repro-2") produced reprocessed tropospheric parameters covering ZTD data spanning 1994 to 2013 at about 300 stations in the IGS network. Beyond IGS, other institutes, such as the Geodetic Observatory Pecný (GOP), have conducted similar efforts. GOP, for example, reprocessed GNSS data at stations in the Regional Reference Frame
- sub-commission for Europe Permanent Network (EPN) from 1996 (30 sites) to 2014 (300 sites) (Dousa et al., 2017), producing a combined ZTD dataset for EPN stations using data from five analysis centres (Pacione et al., 2017). From another aspect, although an enhanced integrated water vapour dataset from more than 10000 global GNSS stations was determined in (Yuan et al., 2023), the dataset is limited only to the year 2020. Therefore, while these reprocessed GNSS datasets provide valuable insights into trends and variations in water vapour, their utility is constrained by the relatively low site density and inadequate
- temporal coverage, necessitating further expansion and extended data acquisition endeavours. In this work, we reprocessed historical GNSS observations from over 5000 stations, covering a 22-year period 2000–2021. The goal is to fulfil the requirements of climate studies for homogeneous, long-term atmospheric parameters across a broad network. This reprocessing campaign, named "GPAC-Repro" hereinafter, used precise satellite orbit, clock, and EOP products from the third IGS data reprocessing campaign (IGS Repro-3), in conjunction with state-of-the-art strategies and models to
- further ensure the quality and consistency of the dataset. The ZTD estimates derived from the GNSS data were converted to PWV using temperature and pressure data from the fifth generation of European ReAnalysis (ERA5) atmospheric reanalysis (Hersbach et al., 2020). Then, a rigorous quality assessment of the determined ZTD and PWV estimates was conducted by comparing them with their counterparts from ERA5, radiosonde, and Very Long-Baseline Interferometry (VLBI). Additionally, to elucidate the characteristics of the new dataset and facilitate its use in climate studies, we also calculated the maximum and
- minimum, as well as daily, monthly, and annual mean values of PWV and ZTD for each station over the entire study period. Overall, this newly reprocessed, long-term, homogeneous GNSS climate dataset is one of the most comprehensive GNSS atmospheric datasets available. It represents a significant advancement in the innovative field of GNSS climatology, providing a valuable resource for scientific communities engaged in climate studies.

#### 2 Data and Methods

# 120 2.1 Data acquisition and analysis

This reprocessing campaign initially utilised GNSS observations from 5180 globally distributed stations, covering a 22-year period 2000–2021. The GNSS data were sourced from four archive centres, including the Crustal Dynamics Data Information System (CDDIS, <a href="http://gdc.cddis.eosdis.nasa.gov/gnss/data/daily">http://gdc.cddis.eosdis.nasa.gov/gnss/data/daily</a>), the Scripps Orbit and Permanent Array Centre (SOPAC,

http://garner.ucsd.edu/pub/rinex), Geoscience Australia (GA, sftp.data.gnss.ga.gov.au), and the Hong Kong Geodetic Survey
 Section of the Survey and Mapping Office (SMO, <u>ftp://ftp.geodetic.gov.hk/rinex2</u>). The daily GNSS observations were stored in the standard Receiver INdependent EXchange (RINEX) format, which contains dual-frequency carrier phase and code measurements, typically recorded at a 30-second sampling interval. Following a rigorous data screening process, 95 sites were excluded due to identified issues with the atmospheric results, leading to a final dataset comprising 5085 GNSS stations. The detailed exclusion criteria and screening procedures are described in Section 3. Fig.1 illustrates the geographical distribution

of the GNSS stations included in the GPAC-Repro campaign, all of which successfully passed the quality control checks. In addition to the distribution, further analysis of the data record duration and integrity across the 5085 sites is presented in Fig.2.

Figure 1. Geographical distribution of 5085 GNSS sites (a). Zoomed-in figures of regions with high station density, including the United States (b), Europe (c), Australia (d), and Japan (e).