# Peer review of "A comprehensive 22-year global GNSS climate data record from 5085 stations"

_Earth System Science Data, 2025_

## Referee Comment (RC2)

This study presents a highly valuable global GNSS climate data record derived from long-term GPS observations. The manuscript is well-structured, and the dataset is of substantial relevance to the meteorological, climate, and even geodetic communities. However, several aspects would benefit from clarification and technical refinement to enhance reproducibility and overall clarity. I would like to recommend the manuscript for publication after minor revisions addressing the following points:

(1) The current title, "A comprehensive 22-year global GNSS climate data record from 5085 stations," may lead readers to assume that each of the 5085 stations have a continuous 22-year time series. To better reflect the actual structure of the dataset, I would like to suggest refining the title by including a phrase such as "covering up to 22 years" or "spanning up to 22 years". Similarly, the first sentence of the abstract and other related statements should be revised to accurately characterize the temporal extent and variability of the dataset across stations.

(2) Line 11: The acronym GPAC is introduced in the manuscript without explanation. Please define this acronym upon its first use, whether it refers to your team, processing framework, or another entity, to ensure clarity for readers. Also, consider removing those unnecessary abbreviations in the abstract to maintain focus and readability.

(3) Line 50: Given the importance of this methodological transition in the context of GNSS atmospheric monitoring, I suggest introducing a paragraph break. This will improve the logical flow by separating the historical context from the introduction of GNSS techniques.

(4) Line 62: The current categorization of detection models may be misleading. AI-empowered methods, while modern, are still broadly encompassed within statistical methodologies. I recommend grouping the models into two categories of statistical and numerical and mentioning AI-empowered methods as a subcategory under statistical models for clarity and conceptual consistency.

(5) Line 145: The manuscript indicates the use of the Bernese GNSS Software (V5.2) for data processing, but no reference is provided. Please include an or a few appropriate citations to support reproducibility.

(6) Table 1: Considering that the newer VMF3 has been released and is widely reported to offer improved accuracy over VMF1, please clarify the rationale for using VMF1 in this study.

(7) Table 1: The study relies on reprocessed orbit and clock products from the CODE analysis center. Given that several other IGS analysis centers also produce high-quality reprocessed solutions, it would strengthen the manuscript to briefly explain why CODE products were preferred for this work.

(8) L
    i

(9) How did you handle the station coordinates in your processing? Were the coordinates estimated simultaneously along with the tropospheric parameters and other parameters, or were a priori coordinates introduced and kept fixed during the GNSS data processing?

(10) Lines 177 and 190: Some units for physical constants are presented in a format inconsistent with SI or typographic standards. Please carefully review the entire manuscript to ensure uniformity in unit notation, including consistent use of spaces between values and units.

(11): Figures 7, 12 and 14: While the figures in the manuscript are generally informative and diverse, the legends and axis labels in Figures 7, 12, and 14 are too small and difficult to
    T
    h
    e

    u

read. Please consider increasing font sizes and improving color contrasts to enhance readability, especially for printed versions.

(12) Line 521: When discussing monthly variations in PWV, it would be helpful to mention the number of hourly samples per month (e.g., range from A to B), as the number of days in a month varies and this affects statistical interpretations.

(13) Figure 19: The current subfigure titles in Figure 19 are somewhat generic, with panels (a) and (c) and panels (b) and (d) labeled identically. I would like to suggest updating the titles to more clearly differentiate between metrics and highlight their specific content.

(14) Line 624: As a gentle reminder, if the DOI for the dataset deposited in the PANGAEA data repository is available, please directly include it in the revised version. This will improve accessibility and ensure the completeness of your data publication.

---

## Author Comment (AC1)

**General Comments**

1. Overall, I find that the dataset is of general interest to the community and should be published. The first four sections are of the highest priority, whereas the section on "Further Analysis" more describes applications for future users of the dataset. Therefore, my attention is focused on the first sections to have a clear and understandable description in order to increase the possibility of many users of (and citations to) the dataset.

**Response**: We are sincerely grateful for your thorough review and constructive comments. Our point-by-point responses to each comment are provided below.

2. I have a small problem with the title because it gave me the impression that all 5085 stations have 22-year long time series. In fact, the majority of the stations have time series that are significantly shorter. What about something like: "A global GPS climate data record for 5085 stations covering up to 22 years"?

**Response**: We totally agree with your constructive suggestion, and the original title could indeed lead to unnecessary misleading and misunderstanding regarding the length of the dataset. To avoid this ambiguity while remaining faithful to the scope of the manuscript, we revise the title to:

*"A global GNSS climate data record from 5085 stations spanning up to 22 years".*

It is worth noting that we also note your subsequent comment regarding the utilisation of "GNSS" vs. "GPS": (1) regarding the revised title, we intentionally use GNSS to make it a general term, to support future applications of GNSS meteorology and climatology, and to align with the ongoing deployment of multi-constellation, multi-frequency GNSS (which is also our current work); (2) in the main text, we have standardised terminology, using 'GNSS' for multi-constellation content and reserving "GPS" only for GPS-specific contexts (see the responses below).

In addition, we have reviewed the whole manuscript to remove wording that might imply uniform 22-year coverage (like in the Abstract, Introduction, and Section 2.1), and we consistently state that the record lengths vary, using the phrase "*spanning up to a 22-year period*" throughout.

3. Early in the manuscript you make clear that only GPS data are used. Later it is however very often you refer to GNSS data. I think it is okey only when you refer to GNSS data in general terms, not when you discuss your dataset. Furthermore, in the summary section you may have some ideas to discuss the future use of GNSS (rather than just GPS), pros, and cons?

**Response**: We highly value this suggestion. In the revised manuscript,

(1) we have clearly separated contexts where GNSS is appropriate from those where GPS must be used. Specifically, as suggested, when discussing the generated dataset, detailed processing, results, and other GPS-specific contexts, we now use "GPS" consistently. We reserve the use of "GNSS" for those general statements about the field and future directions, primarily in the Introduction and Summary.

(2) In addition, since the dataset was generated using only GPS data, we have also added a few sentences to Section 7, outlining the limitations of the work and our planned new reprocessing campaign, which will incorporate multi-GNSS observations and expand the network through collaboration with regional data centres. We also briefly summarise some potential benefits

(improved satellite visibility and geometry, better temporal availability and robustness) and trade-offs (management of inter-system/inter-frequency biases, harmonisation of antenna calibrations and metadata, and ensuring cross-system consistency):

**Lines 690-697:**

"*Despite these advancements, several key challenges and opportunities for improvement remain. First, while this study mainly employed GPS observations, integrating multi-GNSS systems such as Galileo, GLONASS, and BeiDou could improve satellite visibility and geometry, and may enhance spatiotemporal availability and robustness, particularly in under-represented regions like polar areas and oceans. However, as noted in Section 2.2, introducing additional constellations can impose inter-system biases and calibration complexities that may induce shifts in the time series. In other words, the net benefit is context-dependent and not yet settled. Given this, our ongoing research is conducting a new reprocessing campaign that will incorporate multi-GNSS observations using the latest Bernese V5.4 and updated tropospheric models like VMF3, while managing inter-system and inter-frequency biases, harmonising antenna calibrations and metadata, and ensuring cross-system consistency.*"

**Specific Comments**

1. line 79: You state that a shortcoming of previous studies is that the length of the time series is only around 10 years. This is certainly true and climate scientists often use 30 year averages which means that the time series in your dataset suffer from the same shortcoming. In fact, many of the sites in the dataset have time series of about 10 years or less. I think it is fair to state this. The decision to use 30-year averaging periods was taken by IMO (now WMO) at a congress in 1935. Also, at line 106 you give the false impression that all time series are 22 years.

**Response**: Many thanks for your suggestion, we have made some modifications accordingly:

(1) In Section 1, we have added some sentences and two references to further elaborate on this:

**Lines 77-82:**

"*However, despite recent advances, the potential of GNSS atmospheric monitoring remains largely under-utilised in the climate community. **This is primarily due to the lack of robust long-term GNSS climate datasets, whereas climate applications typically require the use of datasets spanning several decades (e.g., 30-year climatological normal; WMO, 2007; Arguez and Vose, 2011). In this context, many datasets used in the aforementioned studies span only around 10 years, even if this is partly because the limited record length at many sites, such durations is still insufficient for uncovering the climate change signals embedded in these parameters.***"

(2) Please note that, as these statements are used to provide a review of previous work, so we do not refer to our dataset. Instead, we have also added several sentences in Section 7 to state the same objective limitation for our dataset:

**Lines 697-701:**

"*Second, although the generated dataset covers up to 22 years and can support climate applications, **it does not yet meet the "30-year" timescale typically required for climatology**, largely because most GNSS stations lack sufficiently long historical observations as stated earlier. **Accordingly, we aim to continuously process new data***

*streets to extend the record beyond 30 years and provide a more complete dataset.*"

(3) As indicated in our response to the General Comment#2, we have revised wording that could suggest a uniform 22-year coverage, changing "covering a 22-year period 2000–2021" to "spanning up to a 22-year period 2000–2021".

***References used here:***

*[1] Arguez, A., and Vose, R. S.: The definition of the standard WMO climate normal: The key to deriving alternative climate normals. Bulletin of the American Meteorological Society, 92(6), 699-704, doi:10.2307/26218540, 2011.*

*[2] WMO: The role of climatological normals in a changing climate (WMO/TD-No. 1377, WCDMP-No. 61), World Meteorological Organization, Geneva, Switzerland, 2007.*
* * *
**2. line 89:** later in the manuscript you also mention changes of hardware, such as antennas and radomes. It can be meaningful to mention these also here.

**Response**: This issue has been amended in the revised manuscript:

**Lines 92-94:**

"*However, the determined ZTD time series may still exhibit inhomogeneities due to updates to reference frames and models, variations in mapping function implementations, adjustments to elevation cut-off angles, modifications to processing strategies, **and changes in hardware (such as antennas and radomes)**.*"
* * *
**3. line 127:** "Following a rigorous data screening process, 95 sites were excluded due to identified issues with the atmospheric results, leading to a final dataset comprising 5085 GNSS stations." It will be helpful to list these 95 sites, not in the manuscript but in the data archive, together with the reason for excluding the site. I searched for a specific IGS site, with a long time series, but could not find it among the 5085 sites included. Perhaps it is among the 95 sites? In any case it will be helpful for future users of GNSS PWV to be aware of problem sites.

**Response**: We are sincerely grateful for this insightful suggestion. Documenting the excluded sites (and the reasons) will indeed improve the utility of the archive. We are not certain which IGS site you searched, but my gut feeling is that it may be among the 95 excluded sites.

As noted in the manuscript, the dataset is hosted in two places: (1) the PANGAEA repository and (2) our online portal. As per your suggestion, we will first compile a concise list of the excluded sites (including site code, coordinates, and a brief reason for exclusion). Then, for the PANGAEA repository, we will reach out to the Editor and add the list to the archive; for our online portal, we will directly add a specific subsection for easy reference.
* * *
**4. line 162:** Please motivate why you chose such a low cutoff angle. For example, when searching for trends varying systematic errors are very important and should be reduced. Multipath is an effect which get worse closer to the horizon, and this is especially true if the horizon mask change over long time.

**Response**: Many thanks for raising this concern. First, we fully acknowledge that observations at low elevation angles are generally more susceptible to multipath and increased noise, particularly if the horizon mask changes over a longer period. **Our choice of a 3° cut-off elevation angle was made cautiously and is supported by multiple previous literature showing that, when appropriately weighted and modelled, including low-elevation data reduces the correlation**

**between tropospheric parameters and station height and improves the accuracy and stability of long-term ZTD estimates.** More specific justifications are summarised as follows:

(1) In GNSS data analysis, the ZTD, station height, and receiver clock parameters are proven to be strongly correlated, which can introduce significant errors into coordinate and atmospheric estimates (Rothacher et al., 1998). **Including low-elevation observations improves satellite geometry and reduces this parameter coupling.** The Bernese Software manual (Dach et al., 2015, Section 12.4) recommends using a cut-off angle **not larger than 10°** for tropospheric parameter estimation. In addition, Dousa et al. (2017) further showed that, within the second European GNSS reprocessing campaign (1996-2014) aimed at identifying the optimal strategy for estimating tropospheric parameters, **lowering the elevation cut-off angle from 10°/7° to 3° reduces parameter correlations, improves station height repeatability, and enhances accuracy, thus supporting the inclusion of low-elevation observations in long-term GNSS reprocessing**.

**(2)** Minimising the coupling between height and tropospheric parameters is particularly important for climate applications, as height variations can otherwise propagate into ZTD estimates and introduce spurious trends or seasonal signals. This explains why **recent studies report better agreement of long-term GNSS-ZTD trends with independent references when lower cut-offs are used (Bai et al., 2023)**. While Ning and Elgered (2012) noted that GNSS-ZTD trends yield better agreement with radiosonde trends at a higher cut-off angle (~25°), Bai et al. (2023) indicated that this largely stemmed from the use of unhomogenised radiosonde data. **When homogenised radiosonde trends developed by (Dai et al., 2011) or ERA5-derived trends are used, lower cut-off angles (3°–7°) yield improved consistency.**

(3) Another reason for adopting a cutoff of 3° is to **ensure consistency with the reprocessing strategy of CODE in the IGS Repro3 campaign (Dach et al., 2021), whose reprocessed orbits, clocks, ERPs, and ionosphere products are key inputs to our processing campaign**. Although low-elevation data are more susceptible to multipath and have higher noise as illustrated, their impact can be mitigated using elevation-dependent weighting, latest tropospheric models (e.g., VMF1), and strict pre-/post-processing quality-control procedures in Bernese.

To further explain the rationale behind this choice, the following statement has been added:

**Lines 166-171:**

> "*Please note that the use of a low cut-off elevation angle of 3° is based on findings from previous literature, which indicated that including low-elevation observations reduces the correlation between tropospheric parameters and station height, thereby improving the accuracy of ZTD estimates and the reliability of long-term trends (Dach et al., 2015; Dousa et al., 2017; Bai et al., 2023). Moreover, this choice also ensures consistency with strategy adopted by CODE in the IGS Repro-3, whose orbit, clock, and ERP products are used in our study as illustrated earlier (Dach et al., 2021).*"

*References used here:*

*[1] Bai, J., Lou, Y., Zhang, W., Zhou, Y., Zhang, Z., Shi, C., and Liu, J.: Impact analysis of processing strategies for long-term GPS zenith tropospheric delay (ZTD). Atmospheric Measurement Techniques, 16(21), 5249-5259, doi:10.5194/amt-16-5249-2023, 2023.*

*[2] Dach, R., Lutz, S., Walser, P., and Fridez, P.: Bernese GNSS software version 5.2. Astronomical*

*Institute, University of Bern, 858, 2015.*

*[3] Dach, R., Selmke, I., Villiger, A., Arnold, D., Prange, L., Schaer, S., Sidorov, D., Stebler, P., Jäggi, A., and Hugentobler, U.: Review of recent GNSS modelling improvements based on CODEs Repro3 contribution. Advances in space research, 68(3), 1263-1280, doi:10.1016/j.asr.2021.04.046, 2021.*

*[4] Dai, A., Wang, J., Thorne, P. W., Parker, D. E., Haimberger, L., and Wang, X. L.: A new approach to homogenize daily radiosonde humidity data. Journal of Climate, 24(4), 965-991, doi: 10.1175/2010JCLI3816.1, 2011.*

*[4] Dousa, J., Vaclavovic, P., and Elias, M.: Tropospheric products of the second GOP European GNSS reprocessing (1996–2014). Atmospheric Measurement Techniques, 10(9), 3589-3607, doi: 10.5194/amt-10-3589-2017, 2017.*

*[6] Ning, T., and Elgered, G.: Trends in the atmospheric water vapor content from ground-based GPS: The impact of the elevation cutoff angle. IEEE Journal of Selected Topics in Applied Earth Observations and Remote Sensing, 5(3), 744-751, doi: 10.1109/JSTARS.2012.2191392, 2012.*

*[7] Rothacher, M., Springer, T. A., Schaer, S., and Beutler, G.: Processing strategies for regional GPS networks. In Advances in Positioning and Reference Frames: IAG Scientific Assembly Rio de Janeiro, Brazil, September 3–9, 1997 (pp. 93-100). Berlin, Heidelberg: Springer Berlin Heidelberg, 1998.*
* * *
5. line 165: "a 27-hour time window was adopted," How was these 27 hours defined? In the paper by Dousa et al. (which you refer to) I do not find this specific value, only that three days were combined and thereafter the atmospheric estimates from the day in the middle were selected.

**Response**: Many thanks for pointing this out. We have further clarified in the revised manuscript how the 27-hour window is defined and add a new reference here. Specifically, we state that this definition follows the strategy of the United States Naval Observatory (USNO) used in generating the IGS Final Product, which generates daily normal equations with an overlapping window to improve the stability of estimates near day boundaries (Byram, 2017). It effectively reduces edge effects around day boundaries and improves the continuity of the ZTD time series. We have also adjusted the surrounding text to avoid suggesting that this originates from (Dousa et al. 2017).

**Lines 174-179:**

> "*To address this, we adopted a 27-hour time window, that comprises 24 hours from the current day and an additional 3 hours from the subsequent day. This is consistent with the strategy of the United States Naval Observatory (USNO) for the IGS Final Troposphere Product, which forms daily normal equations and improves the stability of tropospheric estimates near day boundaries (Byram, 2017). These equations were subsequently combined across three consecutive days to produce a 3-day solution (Dousa et al., 2017), from which ZTD estimates for the central date were extracted, thereby enhancing the continuity and accuracy of the dataset*"

***References used here:***

*[1] Byram, S.: IGS Final Troposphere Product Update. United States Naval Observatory, Washington DC, USA. https://files.igs.org/pub/resource/pubs/workshop/2017/W2017-PS04-03%20-%20Byram.pdf, 2017*
* * *
6. line 175+: Why did you choose this rather complicated procedure to derive the ZHD. Other studies I have seen use the Saastamoinen model for the gravity with the ground pressure and the site position as input parameters which I have assumed is sufficiently accurate for the ZWD retrieval. Did you examine if there were any differences that motivate your choice?

**Response**: Thanks for your comment, here are our further explanations:

The Saastamoinen model for ZHD is based on the assumption of hydrostatic equilibrium and has

long been widely used because of its simplicity and reliability. When accurate surface pressure is available, it can reach sub-millimetre uncertainty in ZHD and is indeed often adequate for many meteorological or real-time applications. However, recent studies have identified some limitations. For example, Feng et al. (2020) showed that the Saastamoinen model tends to overestimate ZHD in dry regions such as the Antarctic coast and exhibits seasonal biases over continental areas, particularly at high elevations, with typical errors of 2–5 mm (corresponding to about 0.3–0.8 mm in PWV) and a mean RMS error of ~1.7 mm when compared against radiosonde ZHD estimates.

Given that the main aim of this study is to generate a high-quality dataset suitable for long-term climate monitoring, which demands the highest possible accuracy, we then chose a more rigorous method using ERA5 pressure level fields to derive ZHD estimates. It reduces potential systematic biases that could otherwise propagate into the PWV values. This is particularly important where PWV values per se can be only a few millimetres, so a sub-millimetre error may represent a significant fraction, affecting trend analysis and the detection of climatic anomalies. Our choice is therefore informed by previous studies and the underlying theory.

***References used here:***

*[1] Feng, P., Li, F., Yan, J., Zhang, F., and Barriot, J. P.: Assessment of the accuracy of the Saastamoinen model and VMF1/VMF3 mapping functions with respect to ray-tracing from radiosonde data in the framework of GNSS meteorology. Remote Sensing, 12(20), 3337, doi:10.3390/rs12203337, 2020*
* * *
**7. Section 3.3: Why do you wait until this stage by removing unrealistic negative values of the PWV. You do not need reference data for this action.**

**Response**: In this processing campaign, our quality control mainly comprised three stages:

(1) daily-solution screening based on coordinate repeatability (Section 3.1)

(2) ZTD outlier detection using temporal variability and formal errors (Section 3.2)

(3) PWV screening (i.e., **Section 3.3**)

It can be found from the consecutive stages that, only those ZTDs that passed both Stages 1 and 2 were converted to PWV, after which any negative PWV values were flagged and removed in Stage 3. This is to ensure that the removal of negative PWVs was applied to a reliable ZTD time series (after ZTD screening), avoiding spurious removals caused by transient ZTD outliers or coordinate instabilities upstream. In other words, the removal of unrealistic negative PWVs, in fact, does not rely on ERA5 and was performed before the ERA5 comparison in Stage 3. This also explains why this part is described at the beginning, i.e., the first paragraph of Section 3.3.

To make this sequence explicit, we have added a clarifying sentence at the beginning of Section 3.3 noting that negative PWVs are excluded after Stages 1/2 but prior to the ERA5 comparison:

**Lines 293-295:**

"*In the final phase, the screened ZTD estimates were converted to PWV values and further validated using ERA5-derived PWVs as a reference. **Before comparing with ERA5 dataset, we first performed an initial range check to remove unrealistic negative PWV values. The initial check excluded 0.16 % of the estimates.***"
* * *
**8. When you remove outliers by comparing with reference data you assume that the reference data are correct. I think that needs to be discussed. From my point of view, one application of**

GPS/GNSS PWV is to use it as an independent dataset in order to identify problems in other datasets, such as the ERA5.

**Response**: Thanks for putting forward this valuable comment. First, we fully acknowledge that although ERA5-PWV is often used as a reference for quality control in previous studies, doing so implicitly assumes that ERA5 is sufficiently accurate, which may not always hold, particularly in regions with sparse observational coverage, complex topography, or rapidly varying atmospheric conditions. Second, we agree that GNSS-PWV should also be taken as an independent dataset for assessing reanalyses such as ERA5.

To address this and accommodate other use cases, we now provide two versions of the final PWV dataset (supplied to PANGAEA and uploaded to our online portal. *Note that the new datasets will become publicly available after the completion of the curation/review process, which may take a short while depending on the queue*):

(1) an "unfiltered" dataset containing all GNSS-PWV estimates after internal quality control only, without any ERA5-based outlier exclusion

(2) an ERA5-screened/filtered dataset in which ERA5 is used to remove outliers as described in Section 3.3.

Providing both datasets allows users to select the version most appropriate for their applications. For example, for studies that aim to use GNSS-PWV as an independent dataset benchmark against ERA5, we recommend the unfiltered version.

A corresponding clarification has been added to Section 3.3 of the revised version as follows:

**Lines 325-331:**

*"Additionally, it is crucial to note that although ERA5-derived PWV has been widely used as a reference in ZTD/PWV quality control, this practice implicitly assumes that the ERA5 dataset is sufficiently accurate, which may not always hold in all regions, especially where observational constraints are limited or atmospheric variability is high. To accommodate various use cases, we provide two versions of the PWV dataset: an unfiltered product that contains all GNSS-derived PWV estimates after internal quality control, and an ERA5-screened product in which ERA5 is used only to flag and optionally remove gross outliers."*
* * *
9. Table 2: The observing periods of the VLBI stations are much longer. I understand that you do not pick up data before the start of the GPS time series but there should be data after 2018? Please explain.

**Response**: In this study, we used the IVS-combined tropospheric products generated by the GFZ Potsdam Troposphere Combination Centre, which combine tropospheric delay estimates from multiple IVS Analysis Centres. However, the publicly available products currently extend only to the end of 2018 (see the archive at CDDIS: https://cddis.nasa.gov/archive/vlbi/ivsproducts/trop/). Accordingly, our comparison between GNSS and VLBI was only limited to the period up to 2018, consistent with the temporal availability of the IVS-combined data.

As indicated above, we are still working on processing GNSS atmospheric parameters using the latest version of Bernese and to incorporate multi-GNSS observations. Once combined products beyond 2018 become available, we will revisit and extend the comparison and include the results into our future work.

**10. line 420: How do you define a robust agreement? For which value of the STD is it no longer robust?**

**Response**: Thanks for raising this issue. We agree that the term "robust agreement" is a bit vague without a detailed explanation. We have therefore refined the statement from two aspects:

(1) We added a new reference to justify the "3 mm" criterion, which aligns with a commonly used PWV accuracy threshold for climate applications (Offiler et al., 2010).

(2) We also replaced "robust" with "close" to be more moderate and avoid ambiguity.

Accordingly, the statement now reads:

**Lines 454-456:**

> "*Notably, 88.06 % of the sites exhibit mean differences within the range of [-1, 1] mm, and 90.80 % have STD below 3 mm,* **with 3 mm being a commonly used threshold for PWV accuracy in climate applications (Offiler et al., 2010), demonstrating close agreement between the two sets of PWV.***"*

*References used here:*

*[1] Offiler, D., Jones, J., Bennit, G., and Vedel, H.: EIG EUMETNET GNSS Water Vapour Programme (E-GVAP-II). Product Requirements Document, MetOffice, http://egvap.dmi.dk/support/formats/egvap_prd_v10.pdf, 2010*

**11. Section 4.3. Is not clear to me what was the action taken after finding these changepoints. Did you modify the PWV time series in the data archive, or not? Also in this case, are you sure that some detections of changepoints are not due to problems with ERA5? A related question is if you searched for changepoints in the ZHD. If such exist, they will indirectly cause a corresponding jump/offset in the PWV.**

**Response**: In this study, changepoints in the PWV series were identified and reported as metadata flags, and no adjustments were applied to the archived PWV time series. We chose not to modify the series for two reasons: First, to preserve traceability and avoid introducing model dependence, given this is a data description paper. Second, attributing a detected break to a specific source such as data processing, hardware, local environment, or the reference dataset, is non-trivial in a global product, given the currently available information.

Using a reanalysis, such as ERA5, as a reference for changepoint detection is a common practice, typically applied to difference series to reduce natural variability and improve break detectability (Ning et al., 2016; Van Malderen et al., 2020). Benchmarking/sensitivity studies further highlight method dependence and practical caveats, including tests on synthetic or reprocessed GNSS-PWV and dedicated segmentation tools (Nguyen et al., 2021; Quarello et al., 2022). Moreover, documented biases and representativeness differences in reanalyses and GNSS indicate that detections should be interpreted with caution, rather than taken as proof of an ERA5/GNSS error (Bock and Parracho, 2019; Zhang et al., 2019; Van Malderen et al., 2020; Yuan et al., 2025).

In addition, in this study, we did not run a separate changepoint search on ZHD values. However, we mention this explicitly in the revised version and keep the detected breaks as flags for users to apply their homogenisation strategies if needed. To make our intent clear, we add this in the paper:

**Lines 515-520:**

> "**For clarity, the detected changepoints are provided as flags alongside the PWV series,**

> *and the archived PWV time series are not modified based on these detections. Although ERA5 is used as a reference to aid detection, it is not the actual "truth", as previous studies suggest that both reanalysis and GNSS data may contain inhomogeneities (Bock and Parracho, 2019; Zhang et al., 2019; Yuan et al., 2025). Moreover, in this study, we did not perform a separate changepoint search on ZTD, ZHD or Tm. Since PWV is derived from these parameters, any discontinuity in them can induce a corresponding offset in PWV, and we will further examine these in detail in future updates.*"

Finally, in this work, we hope to provide a transparent baseline dataset processed with a standard and widely used method. We encourage the community to apply new/advanced homogenisation and changepoint-detection methods and to share derived products tailored to specific applications; where appropriate, we will incorporating well-documented improvements in future updates/work.

*References used here:*

*[1] Bock, O., and Parracho, A. C.: Consistency and representativeness of integrated water vapour from ground-based GPS observations and ERA-Interim reanalysis. Atmospheric Chemistry and Physics, 19(14), 9453-9468, doi:10.5194/acp-19-9453-2019, 2019.*

*[2] Ning, T., Wickert, J., Deng, Z., Heise, S., Dick, G., Vey, S., and Schöne, T.: Homogenized time series of the atmospheric water vapor content obtained from the GNSS reprocessed data. Journal of Climate, 29(7), 2443-2456, doi:10.1175/JCLI-D-15-0158.1, 2016*

*[3] Van Malderen, R., Pottiaux, E., Klos, A., et al.: Homogenizing GPS integrated water vapor time series: Benchmarking break detection methods on synthetic data sets. Earth and Space Science, 7(5), e2020EA001121, doi:10.1029/2020EA001121, 2020.*

*[4] Nguyen, K. N., Quarello, A., Bock, O., and Lebarbier, E.: Sensitivity of change-point detection and trend estimates to GNSS IWV time series properties. Atmosphere, 12(9), 1102, doi:10.3390/atmos12091102, 2021.*

*[5] Quarello, A., Bock, O., and Lebarbier, E.: GNSSseg, a statistical method for the segmentation of daily GNSS IWV time series. Remote Sensing, 14(14), 3379, doi:10.3390/rs14143379, 2022.*

*[6] Zhang, Y., Cai, C., Chen, B., and Dai, W.: Consistency evaluation of precipitable water vapor derived from ERA5, ERA-Interim, GNSS, and radiosondes over China. Radio Science, 54(7), 561-571, doi:10.1029/2018RS006789, 2019.*

*[7] Yuan, P., Blewitt, G., Kreemer, C., et al.: A global assessment of diurnal discontinuities in ERA5 tropospheric Zenith Total Delays using 10 Years of GNSS data. Geophysical Research Letters, 52(5), e2024GL113140, doi:10.1029/2024GL113140, 2025.*

**Technical Corrections**

> 1. I find that the font size in all figures is unnecessarily small. The size could in general be say 50-100 % larger in order to improve the readability.

**Response:** Thanks for your suggestions. Almost all the figures have been replotted/further refined in the revised version. Please see the uploaded manuscript.

> 2. line 11: GPAC is not explained

**Response:** Many thanks for pointing this out. GPAC actually refers to our joint research centre. In the revised version, we spell out and define the acronym at first mention in the main text:

**Lines 112-113:**

> "*This reprocessing campaign, **led by the GNSS data processing for Positioning, Atmosphere, and Climate research centre (GPAC) and hereinafter referred to as "GPAC-Repro"**, adopted precise satellite orbit, xxxxxx.*"

Regarding the Abstract, to avoid misleading, we have also refined the sentence in line 11:

**Lines 10-12:**

"*This work presents a comprehensive global GNSS climate data record derived from 5085 stations, spanning up to a 22-year period 2000–2021. The dataset was generated using the state-of-the-art processing methodologies and precise products from the International GNSS Service (IGS) Repro-3 initiative.*"
* * *
3. line 34: please explain "data gaps". Are the gaps temporal or spatial or both?

**Response:** Thank you for raising the comment. We now clarify that the data gaps are both spatial (sparse coverage) and temporal (short or interrupted records). The sentence has been revised to:

**Lines 33-34:**

"*Despite global efforts spanning several decades, considerable **spatial and temporal data gaps** remain in the existing climate observing networks.*"
* * *
4. line 53: changes in signals --> changes in the arrival time of the signals.

**Response:** Amended in the revised version.
* * *
5. line 57: With used together --> When used together

**Response:** Amended in the revised version.
* * *
6. Figure 2: use more different colours of the different symbols, both in the a and in the b graph. The present version is useless to see the differences, only the distribution of sites on the globe is clear. Also, I wonder if "data integrity" is identical to "data completeness" in the summary file downloaded from the archive? If so, use the the same expression at both places.

**Response:** Many thanks for your advice.

First, we have replotted Fig.2 using a higher-contrast and clearer symbol styling, hence differences between categories are easier to discern. It is noted that because data length and data completeness are continuous fields, we still maintain a single-hue gradient (with increased contrast and refined breaks) rather than entirely disparate colours, which is common practice for conveying monotonic variation. The revised figure also improves legibility for regional analyses for future users (longer records and higher completeness are now more apparent at a glance).

[Figure]

Figure 2. Recorded length (a) and data integrity (b) of the generated GNSS climate dataset across the 5085 stations

Second, data integrity was indeed intended to represent data completeness. We now harmonise the terminology across the manuscript (in Sections 2.1, 4.2, and 5.2) to data integrity for consistency.

7. lines 137, 141: Because you use British English spelling for "vapour" it will be consistent to write "colour".

**Response:** Many thanks for your meticulous review. We have gone through the whole manuscript and standardised spellings to British English throughout.

8. lines 139, 303, 309, 406, 420, 427, 468, 492, 499, 607: there shall be a space between value and unit according to SI standards.

**Response:** Thanks for your reminder. We have corrected the spacing between values and units in all instances. In addition, other similar issues in the paper have also been refined accordingly.

9. Table 1: If you include the citations in the strategy column you can shorten the running text significantly.

**Response:** Thanks for your suggestion. We have refined Table 1 to include the citations directly in the column and polished the following text accordingly.

**Table 1. Modelling features and corrections adopted in the GPAC-Repro campaign**

| Item | Strategy |
|---|---|
| Observations | GPS L1 and L2 observations with a 300 s sampling rate |
| Orbit/Clock/ERP | Products from CODE Repro-3 campaign (Dach et al., 2021) |
| Sub-daily EOP model | High frequency pole model (Desai and Sibois, 2016) |
| Gravity field model | EGM2008 up to degree and order 12 (Pavlis et al., 2012) |
| Solid Earth Tides, Solid and Ocean Pole Tides | IERS Conventions 2010 (Petit and Luzum, 2010) |
| Ocean Tide loading | FES2014b ocean tide loading model (Lyard et al., 2006) |
| Atmospheric tides | Not applied |
| Nontidal loadings | Not applied |
| Ionosphere | First-order effect was eliminated by forming the ionosphere-free linear combination, high order ionosphere (HOI) effect was corrected using CODE global ionosphere model |
| Cut-off elevation angle | 3° |
| Antenna model | igsR3_2077 mode for receiver and satellite phase centre offsets and variations |
| Mapping function | VMF1 (Boehm et al., 2006) |
| Priori hydrostatic delay | VMF1 (Boehm et al., 2006) |
| Troposphere gradient models | The Chen-Herring gradient model (Chen and Herring, 1997) |
| Troposphere-estimated parameters | ZTD (1 hour) and horizontal parameters (24 hours) |
| Solution type | Precise Point Positioning (PPP) |
| Data Span | Long-arc solutions include the data from three days, combined on normal equation level, ZTD and gradient parameters are extracted from the middle day |

10. lines 176, 190, 328: Units shall not be in italics.

**Response:** Amended in the revised version.

11. Eq. (4): will be more clear if it is split into two equations. Furthermore, the cosine function shall not be written in italic font.

**Response:** Many thanks for your valuable suggestion, these issues have been amended. Given the renumbering, we have also updated all subsequent equation references throughout the manuscript.

$$g_s \approx 9.80620 \cdot \left(1 - 0.0026442 \cdot \cos(2\varphi) + 5.8 \cdot 10^{-6} \cdot \cos^2(2\varphi)\right) \quad (4)$$

$$R_s = 6378.137/(1.006803 - 0.006706 \cdot \sin^2(\varphi)) \quad (5)$$

12. line 205: improve the contrast between font and background colour.

**Response:** Thanks for your suggestion. This figure has been replotted in the revised version:

[Figure]

Figure 3. Flowchart of the multi-step data screening approach

13. Figure 4: Also, in this figure it is difficult to see the different colours.

**Response:** Thanks for your reminder. This figure has been replotted:

[Figure]

Figure 4. STD and bias in ZTD at 390 pairs of co-located GNSS stations

14. line 239: These stations are referred to as MDO1, MDO2, and MDO3 in Figure 5 (not MGxx).

Furthermore, when you show examples in the manuscript, I think it will be informative if you added where the stations are located. It will save the work of going into the data archive.

**Response:** Thanks for pointing this out, and we truly apologise for making this careless mistake. We have made the following corrections:

(1) After a careful double-check, we found that the station codes in the main text are correct (i.e., MDO1, MGO2, MGO3). However, the labels shown in Figure 5 were incorrect (i.e., MGxx). Given this, we have re-plotted Figure 5 with the correct station codes to ensure its accuracy

[Figure]

Fig. 5 ZTD differences among three pairs of co-location stations in Texas, USA: MDO1–MGO2 (blue), MDO1–MGO3 (red), and MGO2–MGO3 (yellow)

(2) We now state where these stations are located in the main text and the captions. We indicate the region (Texas and Colorado, USA) and provide representative coordinates:

**Lines 262-269:**

*"**As illustrated by three co-located GNSS stations in Texas, USA (30.68° N, 104.01° W) in Fig. 5**, xxxxxx. Another common source of discrepancies, as mentioned before, is errors in recording receiver or antenna types, often due to human errors. **As illustrated by two pairs of co-located GNSS stations, i.e., PUB1 vs. PUB2 and PUB5 vs. PUB6, in Colorado, USA (38.29° N, 104.35° W) in Fig. 6**, xxxxxx."*

15. line 292: "with 95 problematic sites" indicates that these are included among the 5085 sites. Please rewrite.

**Response:** We have rephrased this sentence in the revised manuscript:

**Lines 322-324:**

"*After completing the rigorous multi-step data screening process, the final dataset comprises 435.65M hourly PWV samples from 5085 sites, **i.e., with 95 sites excluded as problematic and 1.09M hourly samples removed as outliers from the initial set of 5180 stations.***"

16. Figure 11: Also, this Figure is difficult to get any useful information from Are all these radiosonde stations used? I mean is there a GPS site close enough. Perhaps increase the size of the graph and make the VLBI symbols larger? Or delete the figure?

**Response:** Many thanks for your feedback. This figure has been removed from the manuscript, as the essential information is conveyed in the accompanying text and subsequent figures. In addition, the figure numbering has been updated throughout accordingly.

17. Figure 14: the quality should be improved so that it is clear where the GPS sites are located. Perhaps these graphs are not needed as well? Everyone interested for sure knows the topography of Hawaii and the Andes.

**Response:** Thank you for the suggestion. After careful consideration, we have removed this figure from the manuscript. We have also updated the figure numbering throughout to reflect this change.

18. lines 406-407: Is this true solar time, local time, or UT?

**Response:** Thank you for pointing this out. The daytime/night-time windows refer to local time. We have revised this statement to:

**Lines 437-438:**

"*each date must include at least one observation during both daytime (08:00–18:00, local time) and nighttime (18:00–20:00, local time).*"

19. line 464: You can add "radomes" here.

**Response:** This sentence has been revised to:

**Lines 499-501:**

"*Despite the fact that GNSS reprocessing eliminates changepoints caused by inconsistencies in data processing strategies, **the determined PWV time series may still include offsets introduced by receiver, antenna or radome replacements, and observation environment changes**.*"

20. line 491: Equator --> equator.

**Response:** Amended in the revised version.

21. Figure 18: Improve the contrast between the symbols, e.g., light green for daily values and black for monthly values.

**Response:** Thanks for your suggestion. The figure has been replotted in the revised version

[Figure]

Figure 18. Time series of daily and monthly mean PWV and ZTD values at the NYAL and PALM sites over the study period

22. Figures 20 and 21: Colour only areas around the GPS sites, say with a radius of 10-20 km?

**Response:** Many thanks for your advice.

First, we do agree that the PWV over a GNSS station is often representative of conditions within a limited footprint, say a radius of 10–20 km as suggested. We also fully recognise that the current maps convey a broad, qualitative picture rather than fine-scale structure. In practice, however, site coverage over the study region (west coast of the US) is uneven and relatively sparse, after testing buffers from 10 km up to 50 km, colouring only the immediate surroundings of stations produced highly fragmented speckled maps that were difficult to interpret at the regional scale. Second, we note that some previous studies, such as (doi:10.5194/essd-15-723-2023, Fig. A1), have also used similar region-scale renderings even when site numbers are limited and their spatial distribution is uneven. For these reasons, we have retained the present rendering to preserve readability at map scale.

To address your concern, we have also made two concrete changes: (1) We now mask ocean areas (as per the other reviewer's suggestion), and (2) add a few sentences in the text to explicitly state the limitations and recommend that fine-scale analyses be carried out over smaller domains with denser networks.

[Figure]

Figure 20. Rendered images of the 16-year climatological monthly mean PWV values for each month, derived from 590 GNSS sites located on the West Coast of the United States.

[Figure]

Figure 21. Rendered images of the annual mean PWV values for each year over the 16-year period 2006–2021, calculated from 590 GNSS sites located on the West Coast of the United States.

**Lines 613-615:**

"*Notwithstanding, this figure is mainly intended for regional-scale visualisation and qualitative analysis, given the uneven station spacing, they should not be interpreted at fine spatial scales.*"

**Lines 635-636:**

"*As with the monthly fields, the rendering is also qualitative at sub-regional scales. Users seeking local detail in future applications should restrict analyses to smaller areas with denser network coverage.*"

23. line 571: Northern Hemisphere --> northern hemisphere.

**Response:** Amended in the revised version.

---

## Author Comment (AC2)

**Main Comments:**

1. This study presents a highly valuable global GNSS climate data record derived from long-term GPS observations. The manuscript is well-structured, and the dataset is of substantial relevance to the meteorological, climate, and even geodetic communities. However, several aspects would benefit from clarification and technical refinement to enhance reproducibility and overall clarity. I would like to recommend the manuscript for publication after minor revisions addressing the following points

**Response**: We appreciate your thorough review and insightful comments. Our point-by-point responses to each comment are provided below.

**Minor Comments:**

1. The current title, "A comprehensive 22-year global GNSS climate data record from 5085 stations," may lead readers to assume that each of the 5085 stations have a continuous 22-year time series. To better reflect the actual structure of the dataset, I would like to suggest refining the title by including a phrase such as "covering up to 22 years" or "spanning up to 22 years". Similarly, the first sentence of the abstract and other related statements should be revised to accurately characterize the temporal extent and variability of the dataset across stations.

**Response**: We totally agree with your constructive suggestion, and the original title could indeed lead to unnecessary misleading and misunderstanding regarding the length of the dataset. To avoid this ambiguity, we have revised the title to:

> *"A global GNSS climate data record from 5085 stations spanning up to 22 years".*

In addition, we have reviewed the whole manuscript to remove wording that might imply uniform 22-year coverage (such as in the Abstract, Introduction, and Section 2.1), and we now consistently state that the record lengths vary, using the phrase "***spanning up to a 22-year period***" throughout.

2. Line 11: The acronym GPAC is introduced in the manuscript without explanation. Please define this acronym upon its first use, whether it refers to your team, processing framework, or another entity, to ensure clarity for readers. Also, consider removing those unnecessary abbreviations in the abstract to maintain focus and readability.

**Response:** Many thanks for pointing these out.

First, GPAC actually represents our joint research centre. In the revised version, we spell out and define the acronym at first mention in the main text:

**Lines 112-113**

> "*This reprocessing campaign, **led by the GNSS data processing for Positioning, Atmosphere, and Climate research centre (GPAC) and hereinafter referred to as "GPAC-Repro"**, adopted precise satellite orbit, xxxxxx*"

Regarding the Abstract, we have removed unnecessary abbreviations and refined these sentences:

**Lines 10-12**

> "*This work presents a comprehensive global GNSS climate data record derived from 5085 stations, spanning up to a 22-year period 2000–2021. The dataset was generated using the state-of-the-art processing methodologies and precise products from the*

*International GNSS Service (IGS) Repro-3 initiative.*"

**Lines 13-16**

"*A rigorous data screening and quality assessment framework was implemented, including formal error detection, offset identification, and extensive cross-validation with ERA5 reanalysis dataset, radiosonde profiles, and Very Long Baseline Interferometry measurements*"
* * *
3. Line 50: Given the importance of this methodological transition in the context of GNSS atmospheric monitoring, I suggest introducing a paragraph break. This will improve the logical flow by separating the historical context from the introduction of GNSS techniques.

**Response:** As per your suggestion, we have separated the paragraph into two. Thanks for helping us improve the quality of the manuscript.
* * *
4. Line 62: The current categorization of detection models may be misleading. AI-empowered methods, while modern, are still broadly encompassed within statistical methodologies. I recommend grouping the models into two categories of statistical and numerical and mentioning AI-empowered methods as a subcategory under statistical models for clarity and conceptual consistency.

**Response:** Thanks for this insightful advice. Basically, AI-empowered approaches can indeed be subsumed within statistical methodologies given their underlying mechanisms. We now revise the text to group detection models into two categories:

**Lines 62-64**

"*In recent years, the innovative utilisation of GNSS-derived ZTD and PWV estimates has spurred the development of* **statistical (including artificial intelligence-empowered) and numerical approaches** *for nowcasting and very short-range forecasting of weather extremes, such as heavy precipitation and tropical cyclones.*"
* * *
5. Line 145: The manuscript indicates the use of the Bernese GNSS Software (V5.2) for data processing, but no reference is provided. Please include an or a few appropriate citations to support reproducibility.

**Response:** Thanks for your reminder. A new reference has been added:

**Lines 151-152**

"*Advanced modelling and correction techniques were implemented using Bernese GNSS Software* **Version 5.2 (Dach et al., 2015)***, incorporating the latest updates to enhance accuracy*"

***References used here:***

*[1] Dach, R., Lutz, S., Walser, P., and Fridez, P.: Bernese GNSS software version 5.2. Astronomical Institute, University of Bern, 858, doi:10.7892/boris.72297, 2015*
* * *
6. Table 1: Considering that the newer VMF3 has been released and is widely reported to offer improved accuracy over VMF1, please clarify the rationale for using VMF1 in this study.

**Response:** Thanks for raising this point. We fully acknowledge that the newer VMF3 generally improves the approximation of ray-traced slant delays, particularly at low elevations, relative to VMF1 (Landskron and Böhm, 2018). However, for this reprocessing campaign (initiated in 2017),

we intentionally adopted VMF1 to ensure homogeneity and reproducibility across 2000–2021 and to remain consistent with the IGS Repro3 inputs and the Bernese software V5.2. VMF1 remains widely used in high-accuracy geodetic processing and provides well-established global grids for the entire study period (Dach et al., 2021). In addition, we also noted in some previous studies that the benefits of VMF3 are nuanced, with mixed impacts on wet-mapping performance relevant to PWV retrieval (Feng et al., 2020).

To further refine this work, as mentioned in Section 7, our planned reprocessing campaign will incorporate multi-GNSS observations using Bernese V5.4 and will evaluate updated tropospheric models such as VMF3 to further improve the quality of the dataset.

**Lines 695-697**

> "*Given this, our ongoing research is conducting a new reprocessing campaign that will incorporate multi-GNSS observations using the latest Bernese V5.4* **and updated tropospheric models like VMF3**, *while managing inter-system and inter-frequency biases, harmonising antenna calibrations and metadata, and ensuring cross-system consistency.*"

***References used here:***

*[1] Dach, R., Selmke, I., Villiger, A., Arnold, D., Prange, L., Schaer, S., Sidorov, D., Stebler, P., Jäggi, A., and Hugentobler, U.: Review of recent GNSS modelling improvements based on CODEs Repro3 contribution. Advances in space research, 68(3), 1263-1280, doi:10.1016/j.asr.2021.04.046, 2021.*

*[2] Feng, P., Li, F., Yan, J., Zhang, F., and Barriot, J. P.: Assessment of the accuracy of the Saastamoinen model and VMF1/VMF3 mapping functions with respect to ray-tracing from radiosonde data in the framework of GNSS meteorology. Remote Sensing, 12(20), 3337, 2020.*

*[3] Landskron, D., and Böhm, J.: VMF3/GPT3: refined discrete and empirical troposphere mapping functions. Journal of geodesy, 92(4), 349-360, 2018.*
* * *
7. Table 1: The study relies on reprocessed orbit and clock products from the CODE analysis center. Given that several other IGS analysis centers also produce high-quality reprocessed solutions, it would strengthen the manuscript to briefly explain why CODE products were preferred for this work.

**Response:** In this work, the reasons for choosing the CODE products are as follows:

First, CODE products provide a complete and consistent suite of inputs for the entire study period, which avoids cross-centre inconsistencies and supports long-term homogeneity. In other words, to keep the dataset homogeneous, we used a single source in this release, despite other IGS Analysis Centres also producing high-quality reprocessed solutions.

Second, the CODE products are aligned with the Bernese software V5.2 and parameterisation, which reduces configuration uncertainty and improves reproducibility.

Third, these products are widely employed and also well-validated within the community, making it a reliable baseline for global climate-scale reprocessing.
* * *
8. How did you handle the station coordinates in your processing? Were the coordinates estimated simultaneously along with the tropospheric parameters and other parameters, or were a priori coordinates introduced and kept fixed during the GNSS data processing?

**Response:** Thanks for pointing this out. In this reprocessing, station coordinates were estimated simultaneously with the tropospheric parameters. A priori coordinates from the weekly combined solutions were taken as initial values and applied as tight constraints, which were not fixed during

the processing. Coordinates were treated as static over each 27-hour session, while receiver clocks were estimated epoch-wise.

9. Lines 177 and 190: Some units for physical constants are presented in a format inconsistent with SI or typographic standards. Please carefully review the entire manuscript to ensure uniformity in unit notation, including consistent use of spaces between values and units.

**Response:** Thanks for your kind reminder. We have gone through the whole manuscript, and all the issues occurred in the original manuscript have now been revised/refined.

10. Figures 7, 12 and 14: While the figures in the manuscript are generally informative and diverse, the legends and axis labels in Figures 7, 12, and 14 are too small and difficult to read. Please consider increasing font sizes and improving color contrasts to enhance readability, especially for printed versions.

**Response:** Thanks for your comment. In the revised version, almost all the figures, including Figs. 7 and 12, have been replotted or further refined to ensure readability. Regarding Fig. 14, as per the other reviewer's suggestion "*Everyone interested for sure knows the topography of Hawaii and the Andes*", we have removed it from the manuscript and updated the figure numbering throughout to reflect this change. Here are the updated figures 7 and 12:

[Figure]

Figure 7. Identification of nearby stations for SNGO (a) and time series of PWV differences with threshold limits (b)

[Figure]

Figure 12. The mean and standard deviation of differences in PWV between GPS and ERA5

11. Line 521: When discussing monthly variations in PWV, it would be helpful to mention the number of hourly samples per month (e.g., range from A to B), as the number of days in a month varies and this affects statistical interpretations.

**Response:** Thanks for your valuable suggestion. We now quantify the completeness thresholds for both daily and monthly periods and add a new sentence in the revised version:

**Lines 565-568**

"*To minimise the impact of missing data on the analysis, we applied a strict inclusion criterion, i.e., only days with at least 21 hourly estimates and months with a minimum of 650 hourly samples were included in the calculation, corresponding to **at least 87.5 % (21 out of 24 hours) daily completeness and 87.4–96.7 % (650 of 672–744 hours) monthly completeness, nominally approximate 90 %**.*"

12. Figure 19: The current subfigure titles in Figure 19 are somewhat generic, with panels (a) and (c) and panels (b) and (d) labeled identically. I would like to suggest updating the titles to more clearly differentiate between metrics and highlight their specific content.

**Response:** Thanks for your valuable suggestion, we have replotted this figure in the revise version:

[Figure]

Figure 19. RMS (a, b) and Bias (c, d) statistics resulting from the comparison of daily and monthly mean PWV at all the stations against ERA5 dataset over the whole study period.
* * *
13. Line 624: As a gentle reminder, if the DOI for the dataset deposited in the PANGAEA data repository is available, please directly include it in the revised version. This will improve accessibility and ensure the completeness of your data publication.

**Response:** Thanks for your reminder. We have updated the Data Availability section in the revised version, in which the DOI of our submitted dataset has been provided:

**Lines 673-675**

*"The global GNSS climate data record, including hourly ZTD and PWV estimates, described in this work is now available at: https://doi.org/10.1594/PANGAEA.982476 (Wang et al., 2025). Additionally, the datasets have also been made accessible at: https://www.gnss.studio/Login, with its data download interface shown in Fig. 22."*

---

## Author Comment (AC3)

**Community Comment**

**General Comments**

1. Thank you for presenting this new GNSS dataset, which holds potential for climate studies. I found the manuscript well-structured and written, and the methodology sound. However, I would like to share a few comments and questions regarding some of the methodological choices and propose suggestions to further enhance the QC/QA process and overall quality of your dataset.

**Response**: Many thanks for your thorough and encouraging review/assessment of our work. We appreciate your comments and suggestions. Our point-by-point responses to each comment are provided below.

**Major Comments**

1. Section 2.2 GNSS data processing: While you mention adhering to the highest standards in your study, it is worth noting that the analysis was conducted using Bernese GNSS Software version 5.2. Since 2022, version 5.4 has been available, introducing several improvements. These include enhanced observation (RINEX) quality control and preprocessing, improved ambiguity resolution for PPP, and updated tropospheric models such as VMF3. Considering these advancements might further strengthen the robustness and quality of your dataset.

**Response:** Many thanks for this constructive suggestion. We prepared the reprocessing campaign in early 2020 and commenced it soon after the release of the IGS Repro3 products, using Bernese V5.2. The end-to-end campaign (data processing followed by QC/QA and validation) spanned several years, and at the outset we had not yet obtained a license for Bernese V5.4. This, in fact, is the main reason Bernese V5.2 was adopted for the present work.

We fully acknowledge the advances in V5.4, including enhanced observation quality control, improved ambiguity resolution for PPP, and updated tropospheric models e.g., VMF3, and agree these can further enhance the robustness and accuracy of tropospheric estimates. Hence, since late 2023, we have been working on processing operational datasets (Final and Rapid ZTD results) using Bernese V5.4 to take advantage of these improvements. In addition, we are also planning a new reprocessing campaign with Bernese V5.4 that will include multi-GNSS observations (this release is GPS-only) and expand network coverage via collaboration with regional data centres.

To address your concern (together with the other reviewers' related comments), we have added a few sentences in Section 7 noting a crucial limitation of this release, i.e., the dataset was produced with Bernese V5.2 and GPS-only data and illustrating our ongoing and planned upgrades.

**Lines 690-697**

"*Despite these advancements, several key challenges and opportunities for improvement remain. **First, while this study mainly employed GPS observations, integrating multi-GNSS systems such as Galileo, GLONASS, and BeiDou could improve satellite visibility and geometry, and may enhance spatiotemporal availability and robustness, particularly in under-represented regions like polar areas and oceans. However, as noted in Section 2.2, introducing additional constellations can impose inter-system biases and calibration complexities that may induce shifts in the time series. In other words, the net benefit is context-dependent and not yet settled. Given this, our ongoing research is conducting a new reprocessing campaign that will incorporate***
* * *
2. Section 2.3 Retrieval of PWV: the calculation of ZHD from the numerical integration of refractivity is not recommended (Jones et al., 2020, chap. 5.4.2), especially when only 37 pressure levels are available such as with ERA5. Instead, Saastamoinen formula should be used with surface pressure which can still be computed from ERA5 (with adequate interpolation between levels). This is the approach actually used by Haase et al. (2003) for their GNSS ZTD to IWV conversion. Note that they use only Eq. (2) for the integration of radiosonde profiles, which have many more vertical levels.
* * *
**Response:** Thanks for pointing this out. ZHD can be obtained either via the Saastamoinen formula using surface pressure or by numerical integration of refractivity from reanalysis profiles. The Saastamoinen formula is reliable where accurate surface pressure is available. However, for GNSS sites without co-located pressure sensors, surface pressure must be interpolated or extrapolated from reanalysis fields (e.g., ERA5), depending on the site's position relative to the lowest pressure level of ERA5. In regions with complex topography, interpolation or extrapolation errors can propagate directly into ZHD when using the Saastamoinen formula, because ZHD is proportional to pressure errors. This is the main reason we adopted the integration method using ERA5 profiles. Specifically, by using multiple atmospheric layers, it reduces sensitivity to single-level pressure mismatches and helps maintain consistency across the large global network. From another aspect, the Saastamoinen formula assumes hydrostatic equilibrium and may not fully characterise vertical atmospheric variability, particularly in regions with complex topography or under severe weather conditions.

While Jones et al. (2020) recommend the Saastamoinen formula over integration approach due to potential integration errors, the "net/actual" impact of the two ZHD pathways on PWV, especially for long-term trends/behaviours, remains insufficiently quantified. To the best of our knowledge, there is no definitive experimental evidence/statistics showing that the Saastamoinen formula with interpolated or extrapolated pressure systematically outperforms ZHD integrated from ERA5's 37-level profiles. In the revised version, we clarify our rationale in Section 2.3 and acknowledge in Section 7 that the choice of ZHD method may influence PWV at some sites and times, while emphasising the need for further study.

**Lines 181-184**

> *"The retrieval of PWV from ZTD requires the inclusion of meteorological parameters, specifically temperature and pressure, at the locations of GNSS sites. **However, the absence of meteorological sensors at most stations presents a significant challenge in obtaining these parameters. To address this and maintain consistency across the global network, this study used atmospheric data from the high-quality ERA5 dataset to provide the necessary meteorological inputs.**"*

**Lines 701-704**

> *"Thirdly, the refinement of data retrieval techniques is necessary to address challenges posed by complex topographies and high-altitude regions, thereby improving robustness in these environments. **In particular, the ZHD estimation choice adopted in this study may influence PWV at some certain sites and times, in our next reprocessing, we will***

*document and benchmark the differences between approaches."*

To support community assessment, we hope to release the data, including ZTD, ZHD, ZWD, PWV, and Tm), enabling users to recompute PWV with a Saastamoinen-based ZHD if preferred. We also plan to implement a side-by-side comparison of Saastamoinen- and integration-based ZHD within our next reprocessing campaign. If this is of interest, we would warmly welcome you (appreciate any guidance from you) and any interested colleagues to join this systematic evaluation.
* * *
3. Section 3.2 Screening based on GNSS-ZTD results only: proper credit should be given to Bock, 2020, who introduced the general approach and the methodology to choose range-check and outlier check limits for GNSS ZTD and formal errors which are actually followed here.
To clarify the purpose and usage of the formal errors for the screening process, it may be judicious to more section 4.1 here.

**Response:** Many thanks for your suggestion.

First, throughout the entire processing campaign over the past few years, we have indeed read a lot of publications from Prof. Bock. The range check and outlier detection framework we follow also builds on (Bock, 2020). In the revised version, we now cite the reference where the screening limits and methodologies are introduced to acknowledge this great contribution.

Second, after careful consideration and our internal discussion, we prefer to stick to the status quo and retain Section 4.1 within Section 4. Our rationale is to preserve a clear separation of roles and improve overall readability.

Specifically, Section 3 (data screening) documents how we process daily-solution screening based on coordinate repeatability (Section 3.1); ZTD outlier detection using temporal variability and formal errors (Section 3.2); and PWV screening (Section 3.3). In short, it presents methodologies and screening criteria. Section 4 (quality assessment) reports the distributions of formal errors and their temporal variation (Section 4.1); cross-comparison of PWV with external references (Section 4.2); and offset detection (Section 4.3). Despite the shared adoption of formal errors, Section 4 is designed to present results and interpretation. Moreover, keeping these statistics together avoids duplication and maintains a coherent narrative flow.

In response to your suggestions, we have made several modifications in Section 3.2 in the revised version. First, we now cite (Bock, 2020) alongside the description of the general screening method and the rationale for selecting range-check and outlier thresholds; Second, we add a sentence with an explicit cross-reference to Section 4.1; Lastly, and also based on your following Comment#12, we include a clarifying sentence on the screening limit:

**Lines 230-235**

*"Following the coordinate repeatability evaluation, ZTD values underwent further screening utilising range checks and outlier detection, **following the standardised approach outlined in (Bock, 2020)**. As the first step, ZTDs outside the range of 1–3 m (Bock et al., 2014) and those with formal errors ($\sigma\_ztd$) exceeding 10 mm were excluded. Please note that **formal errors in the ZTD estimates are an important indicator of the quality of GNSS atmospheric parameters and are therefore widely used in screening. For context, it shows that 99.996% of formal errors are ≤ 10 mm in this work. More details about the analysis of formal errors are provided in Section 4.1.**"*
* * *
4. Line 225 to 260: referring to systematic biases may be misleading here. Referring to the

"consistency of ZTD estimated from collocated GNSS sites" as you do later on (Line 256) seems more precise.

**Response:** Thank you for this valuable clarification. In the revised version, we have gone through the relevant sentences and replaced the misleading phrases throughout:

**Lines 242-243**

*"While this step ensures a refined ZTD dataset for PWV retrieval without requiring external reference models, e.g., ERA5, it still has several limitations, **particularly in detecting inconsistencies within ZTD time series**."*

**Lines 274-276**

*"**Similar inconsistencies (biases exceeding 20 mm) were also identified** at four additional stations (LRA1, UTK1, UTK2, and CLS6) when compared to co-located stations and the ERA5 dataset."*
* * *
5. I have one concern here with the impact of height differences. As you mention on line 238, "discrepancies … are often attributed to height differences". This effect is actually expected. A simple rule of thumb approach predicts bias in ZHD around 10 mm and a few mm from ZWD as well based on a 50 m height difference. Such biases can be avoided by applying a proper vertical correction such as described in Bock et al., 2022. Following this approach can be very valuable in detecting station-specific biases when several nearby stations are available.

**Response:** Thanks for raising this concern. In this work, the ZTD comparison between co-located stations serves only as a "preliminary" screening/diagnostic to flag potential issues, with no sites excluded at this stage on the basis of co-located differences. The final assessment of data quality relies on the comprehensive comparison of GNSS-PWV against ERA5-PWV, which provides an independent external reference.

Regarding height effects, we totally agree that an appropriate vertical correction is valuable when several nearby sites are available and height differences are non-negligible. In this work, however, 93% of the co-located pairs of stations have height differences within 10 m, which corresponds to an expected ZTD difference of less than ~2 mm. This is well below the typical uncertainty of ZTD estimates. In addition, we also note that applying a vertical correction might introduce additional error, particularly under complex atmospheric conditions and in rugged topography. Hence, on this basis, we did not apply a vertical correction at this preliminary step.

To address your concern, we make some revisions in the manuscript: First, we explicitly state that no vertical correction was applied in the co-located station comparison and explain the rationale. Second, we recommend that users working with specific local networks apply vertical normalisation (Bock et al., 2022) where appropriate, or explore other vertical correction methods:

**Lines 255-262**

*"After a detailed evaluation, discrepancies in ZTD between co-located GNSS sites are often attributed to height differences. **It should be noted that no vertical correction was applied in this study, as the co-location comparison serves only as a preliminary step to identify potential problematic stations and to provide a general indication of the quality of ZTD estimates. In addition, in this network, 93% of station pairs have a height difference within 10 m, corresponding to an expected ZTD difference of ~2 mm. A vertical correction procedure may also introduce errors of comparable or larger magnitude under certain conditions like in complex atmospheric conditions or over rugged topography. Nevertheless, for local analyses where height differences are***

*non-negligible, we recommend applying vertical corrections following methods described in (Bock et al., 2022) or exploring alternative approaches."*
* * *
6. Another concern is with the impact of equipment changes which can mask short-term site-specific biases when computed over long periods. This should mentioned here and also underlines the importance of offset detection that is discussed later.

**Response:** Many thanks for the valuable suggestion. In the revised manuscript, we now add a few sentences to note this explicitly and underline the importance of offset detection:

**Lines 277-279**

*"In addition, **equipment changes can introduce offsets, leading to inconsistencies at co-located sites and biasing long-term trend analyses by masking short-term, site-specific effects when statistics are aggregated over long periods.** More details regarding the offset detection procedure adopted in our study is described in Section 4.3."*
* * *
7. Line 235 to 250: the discussion employs the terms "bias, deviations and differences", are these referring to mean values? Please clarify. Separating the mean and standard deviation of differences rather than using the RMS (which mixes both) would also give more insight into the nature of the differences.

**Response:** We appreciate this valuable suggestion. We have double-checked all the terminologies and now report standard deviation and bias separately throughout the revised version.
* * *
8. Line 260: "additional quality control" would better fit here in place of "additional screening".

**Response:** Amended in the revised version:

**Lines 289-291**

*"Therefore, to address these limitations, **additional quality control** of the dataset is crucial. This can be achieved by comparing ZTD values with an independent reference dataset, such as ERA5, to validate and enhance the overall quality of the results"*
* * *
9. Section 3.3 Screening based on comparison with reference PWV data: eliminating the negative IWV values may not be sufficient and may actually be hiding a more general bias between GNSS and ERA5 (possibly with seasonal variation). To avoid this caveat, it is preferable to compare ZTD values (as also recommended in Jones et al., 2020, chap. 5.4.1). Then you would probably notice a bias and decide to remove the flawed stations or find that representativeness errors in ERA5 are the limitation.

**Response:** Thank you for this constructive advice. We agree that removing negative PWV alone is not sufficient and may obscure broader GNSS–ERA5 biases. To address this and accommodate other use cases (also based on the other reviewers' comments), we made the following updates:

First, we now provide two versions of the final dataset (supplied to PANGAEA and uploaded to our online portal. *Note that the new datasets will become publicly available after the completion of the curation/review process, which may take a short while depending on the queue*):

(1) an "unfiltered" dataset containing all GNSS-PWV estimates after internal quality control only, without any ERA5-based outlier exclusion

(2) an ERA5-screened/filtered set in which ERA5 is used to remove outliers, see Section 3.3.

This allows users to select the version most appropriate for their applications, for example, when benchmarking GNSS against ERA5, we recommend the unfiltered "raw" dataset.

Second, we now add a few sentences to advise that ZTD comparisons are preferable for screening as they avoid additional conversion uncertainties and potential GNSS–ERA5 representativeness differences embedded in PWV, as per Jones et al. (2020). As indicated in our previous responses, the released products include ZTD, ZHD, ZWD, and PWV, enabling data users to conduct direct further checks/analyses.

Lastly, a similar ZTD comparison study is also under our consideration for our future work.

In general, as data contributors, these updates strike a balance between ensuring the integrity and accessibility of the dataset and encouraging data users to perform further in-depth and innovative analyses based on these data.

**Lines 327-331**

"*Moreover, while we only adopt PWV for screening, we encourage ZTD-level comparisons in future analyses, as they avoid additional conversion uncertainties and potential GNSS–ERA5 representativeness differences (Jones et al., 2020). Accordingly, to accommodate various use cases, we provide two versions of the PWV dataset: an unfiltered product that contains all GNSS-derived PWV estimates after internal quality control, and an ERA5-screened product in which ERA5 is used only to flag and optionally remove gross outliers.*"
* * *
10. Line 274-292: I don't understand the rationale of comparing GNSS PWV at a target station to ERA5 PWV at nearby stations. Here you mix to types of errors: GNSS vs. ERA5 (different data sources) and target vs. nearby (difference due to distance). If this procedure is inspired by Nguyen et al., 2024, it is actually not relevant here. Please clarify or correct.
* * *
**Response:** Thanks for this comment. We definitely agree that mixing cross-source and cross-site differences would be problematic. However, our main intention, perhaps insufficiently clear in the previous statements, was to use only PWV differences at the target site as the decision variable, and to use nearby stations to stabilise the monthly dispersion estimates, thus avoiding the mixing of error types. In the revised version, we rephrase these statements and add some explanations:

**Lines 303-313**

"*Following the removal of negative PWV values, a robust outlier detection and elimination method was applied. This method comprises two steps: (1) identifying nearby sites and (2) establishing monthly, site-specific thresholds. First, for each station, nearby stations were identified within 2° in latitude and longitude and with a vertical separation less than 500 m. Next, for the target and each nearby station, we computed the differences between the GNSS-PWVs and ERA5-PWVs. For each month, these differences were pooled to estimate the distribution and define the monthly thresholds of the target station using the aforementioned IQR-based method, i.e.,* $Q_1 - 3 \times IQR, Q_3 + 3 \times IQR$]*, where* $IQR = Q_3 - Q_1$*, and $Q_1$ and $Q_3$ represent the 25th and 75th percentiles, respectively. The resulting monthly thresholds were then applied to the PWV time series of the target station to flag and remove outliers. This procedure was applied to all stations, yielding site-specific thresholds that account for local spatiotemporal variability, and repeated iteratively until no additional outliers were identified. This method provides more robust, locally representative thresholds than using only the PWV differences at the target station,*

*which may fail to detect problematic results when large system inconsistencies exist, such as the PUB2 case in Section 3.1."*

11. Figure 8: this PWV difference series is really suspect. It looks like the ERA5 values in your GNSS – ERA5 differences are very close to zero. Please check.

**Response:** Thanks for raising this concern. We have double-checked the computation and plotting of the differences between GNSS-PWV and ERA5-PWV and confirm that the ERA5 estimates are correctly used in the calculation, and there is no unintended zeroing. The "near-zero" phenomenon in this figure, in fact, mainly results from the scale being dominated by an anomalous surge in the GNSS-PWV time series at AC30 during 2018.

To further clarify this, Fig. 1 shows the time series of GNSS-PWV and ERA5-PWV at AC30. Very low PWV values are actually expected during the winter season at this site, given that it is located at a latitude of ~60° and an elevation of ~750 m (see Fig.2 for the picture of the site). ERA5-PWV generally remains below 10 mm. In 2018, however, it can be found in Fig.1 that GNSS-PWV rises rapidly to ~70 mm, well beyond what is climatologically plausible given its location/elevation.

For comparison, Fig. 3 shows results for a nearby site AC79, located ~25 km from AC30 but at lower elevation (~290 m), where GNSS-PWV and ERA5-PWV agree well and PWV typically remains below 30 mm. This comparison supports the interpretation that the very low ERA5-PWV at AC30 is reasonable, while the rapid increase of GNSS-PWV from ~10 mm to ~70 mm at AC30 is abnormal.

Although we have not yet identified the cause of this abnormal variation in GNSS-PWV, the ZTD results at AC30 provided by UNAVCO (Fig. 4) also exhibit a similar anomalous increase.

Accordingly, we have flagged AC30 as problematic in the text and will continue investigating the underlying cause in our future work.

[Figure]

**Fig. 1 PWV time series over AC30**

[Figure]

**Fig. 2 Picture of the AC30 station**

[Figure]

**Fig. 3 PWV time series over AC79**

[Figure]

**Fig 4 ZTD time series provided by UNAVCO**

12. Section 4.1 Formal errors in ZTD estimations: consider moving this Section to Section 3.2. Give the % of the CDF corresponding to a formal error of 10 mm, which is the limit used for the range check in Section 3.2

**Response:** Many thanks for your suggestion.

First, the exact percentage value corresponding to a formal error of 10 mm is 99.996%. We have added the following sentences in the revised version:

**Lines 340-343**

*"The majority of formal errors range between 0.5 mm and 2 mm, peaking at about 1 mm. The cumulative percentage curve (orange line) rises steeply, reaching 90 % at 2 mm and 99.73 % at 5 mm. The mean and median values of these errors are 1.38 mm and 1.23*

*mm, respectively. **Beyond the X-axis range shown in Fig. 9, as indicated in Section 3.2, this curve attains 99.996% at 10 mm.**"*

Second, as suggested, 10 mm is the determined screening limit used in Section 3.2, hence we have added a contextual sentence there for clarity:

**Lines 230-235**

> *"Following the coordinate repeatability evaluation, ZTD values underwent further screening utilising range checks and outlier detection, following the standardised approach outlined in (Bock, 2020). As the first step, ZTDs outside the range of 1–3 m (Bock et al., 2014) and those with formal errors (σ_ztd) exceeding 10 mm were excluded. Please note that formal errors in the ZTD estimates are an important indicator of the quality of GNSS atmospheric parameters and are therefore widely used in screening. **For context, it shows that 99.996% of formal errors are ≤ 10 mm in this work. More details about the analysis of formal errors are provided in Section 4.1.**"*

Regarding the structure of the manuscript, as also explained in our responses to Comment#3, after careful consideration and our internal discussion, we decided to still retain the results and statistics in Section 4 (Quality assessment). This placement preserves a clear division of roles, i.e., Section 3 documents methods and screening criteria and Section 4 illustrates outcomes and interpretation, as well as improves readability and narrative flow. To address your suggestion and aid navigation, Section 3.2 now explicitly cross-references this subsection.
* * *
13. Line 299: "The formal errors of the estimated ZTD are known to play a key role in analysing the quality of GNSS". Although formal errors may help in the QC/QA of GNSS ZTD estimates (Bock, 2020), it is an overstatement to say that they play a key role. Please moderate.

**Response:** Thanks for your reminder. We now revise this sentence to:

**Lines 338-339**

> *"The formal errors of the estimated ZTD are **a useful indicator** for analysing the quality of GNSS atmospheric parameters (Bock et al., 2020)"*
* * *
14. Section 4.2 Cross-comparison of PWV with external references.
Line 232: the description of ERA5 (number of pressure levels, horizontal resolution, etc.) should be given earlier, e.g. in Section 2.3 when ERA5 is first introduced and used.

**Response:** We are grateful for this constructive advice. These descriptions have been moved to Section 2.3 in the revised version.
* * *
15. Line 383 and 405 + Line 229 (Section 3.2): explain why you chose three different collocation limits for these comparisons (GNSS vs GNSS, GNSS vs. VLBI, and GNSS vs. RS).

**Response:** The use of different collocation limits is due to these comparisons have distinct sensing geometries and data-availability constraints, with diverse implications for representativeness error.

First, regarding the two comparisons GNSS vs. GNSS and GNSS vs. VLBI, both techniques sense ZTD at fixed ground-based stations with comparable theory. We therefore apply a tight collocation criterion (≤1 km horizontal and ≤50 m vertical separation) to minimise representativeness errors.

Second, regarding GNSS vs RS, RS-derived ZTD/PWV are determined from vertical atmospheric profiles along a balloon ascent that typically drifts tens of kilometres over 1–2 hours. Imposing a very small (such as ≤1 km) horizontal limit adds little value for representativeness here and would

severely constrain coverage, e.g., only 22 pairs of co-located GNSS and radiosonde stations would remain with such horizontal limit. To balance representativeness with adequate coverage, we use a horizontal limit of 50 km, yielding 402 pairs of co-located GNSS-RS stations globally. Moreover, this threshold is consistent with many previous studies conducting such intercomparisons (often 50–100 km).

In summary, we use a tight limit when both sensors are fixed, zenith-looking instruments, and a broader, yet standard, limit for GNSS–RS to reflect the balloon's moving sampling volume and to maintain adequate spatial coverage.

In the revised version, we add a few sentences in Section 4.2.3 to clearly illustrate this:

**Lines 439-445**

*"It is worth noting that **we adopt a different collocation limit here than for the GNSS–GNSS and GNSS–VLBI comparisons as radiosonde-derived PWV is obtained along a balloon ascent that typically drifts tens of kilometres over 1–2 hours. Imposing a very small horizontal limit would add little value for representativeness and would severely constrain coverage (e.g., with the same horizontal limit, only 22 GNSS–radiosonde pairs would remain). To balance representativeness and sampling, we therefore use a horizontal limit of 50 km.** Under these criteria, we identified 402 GNSS–radiosonde pairs, with the number of paired PWV samples ranging from 888 to 23,749 (with an average of 7283 samples), equivalent to ~10 years of observations per station."*
* * *
16. Units for ZTD and PWV comparisons: use mm for ZTD and kg/m2 for PWV to avoid confusion. I am wondering whether some of the published GNSS vs. VLBI comparisons are not cited in the wrong unit (e.g. large biases and RMS values).

**Response:** Thanks for your suggestion.

We acknowledge that IWV/PWV can be expressed in $kg/m^2$ or mm, as per many WMO guidelines (https://library.wmo.int/records/item/68695-guide-to-instruments-and-methods-of-observation?offset=3) and also some official resources (https://www.meteo.be/en/research-themes/water-vapour/). On account of the unit of mm is widely used in a substantial number of existing publications, we decide to still report PWV in mm in this study to maintain consistency with much of the literature (including those cited in this work) and aid interpretability. To address your concern and ensure clarity, we now state this equivalence explicitly at first mention in Section 2.3.

**Lines 207-208**

*"Note that, in this study, PWV values are reported in mm, numerically equivalent to $kg/m^2$, i.e., 1 mm = 1 $kg/m^2$, for readability and consistency."*

Second, regarding your concern about the units in the GNSS–VLBI citations (if we understand it correctly), we have double-checked all referenced studies. We identified an issue, i.e., the statistics reported by Steigenberger et al. (2007) should be ZWD, instead of ZTD. This has been amended in the revised manuscript:

**Lines 423-424**

*"For example, Steigenberger et al. (2007) reported a **ZWD** bias of 7.2 mm and an RMS of 14.1 mm, ..."*
* * *
17. Section 4.3 Offset detection: my main concern here is that you applied the detection to a

subset of stations only (less than 50% of all your stations). Considering the use of this dataset for climate studies (e.g. trend analysis), this is a serious drawback. Please explain why not all stations have been checked and whether you intend to further complete this.

**Response:** As stated in the manuscript, "*2485 sites with observation periods exceeding 10 years and data missing rates below 20 % were selected*". That is said we applied two eligibility criteria (record length and data integrity) before conducting offset detection. This choice also reflects the requirements of the adopted methodology. Specifically, sufficient length and continuity are needed to robustly distinguish changes from seasonal variability and serially correlated noise. When time series are short or gappy false-positive rates increase and breakpoint magnitudes become poorly constrained, which risks over-flagging and over-correction. Hence, long and relatively complete records are the primary candidates for trend estimation and homogenisation. In addition, a further practical consideration is that documented equipment/firmware changes (from available log files) are uneven across the global network, and sites with sparse metadata reduce our ability to validate breaks and risk spurious classifications

We recognise the importance of broader coverage for climate applications and, as records lengthen and metadata improve, in our ongoing work, we intend to extend the detection to additional sites and evaluate methods better suited to shorter time series.

18. Another couple of questions concern the confidence that can be placed in the PMTred method and the validation of the detected change-points. First, the number of change points with this method seems a little underestimated compared to studies using other methods (Venema et al., 2012; Van Malderen et al., 2020; Nguyen et al., 2021). Second, the validation with GNSS metadata is not robust as some equipment changes may be missing and not only equipment changes are suspected to produce offsets, but also environment changes. In addition, the numerous firmware changes are not expected to have a significant impact and instead lead to accepting many false detections. To overcome these limitations, it may be advisable to cross-compare the results from different detection methods and to implement a more robust validation method, e.g. based on multiple pairwise comparisons (Caussinus and Mestre, 2004; Menne and Williams, 2009; Nguyen et al., 2024).

**Response:** Thank you for these thoughtful suggestions. We agree that both the choice of method and validation strategy can affect the number of detected changepoints.

First, as contributors in a data description paper, our implementation is intentionally conservative. We analyse monthly PWV differences and require strong statistical evidence: a 95% critical value for changes corroborated by metadata and 99.9% for unrecorded changes. Thie approach reduces false positives in short or noisy records but can achieve fewer detections than methods tuned for higher sensitivity.

Second, in this study, metadata are adopted only for corroboration, not as a substitute for statistical evidence. A recorded change is tagged "documented" only if a statistically significant break is also detected. Conversely, metadata entries do not, by themselves, trigger a break. In addition, we also acknowledge that environmental changes may cause offsets and that site logs can be incomplete.

As in the current release, all detected changepoints are provided as flags, and we do not adjust the data. This allows users to apply their homogenisation strategies, while we expand the method and coverage in our subsequent versions/updates.

To address your concern, we have added these sentences in the revised version:

*"For clarity, the detected changepoints are provided as flags alongside the PWV series, and the archived PWV time series are not modified based on these detections. Although ERA5 is used as a reference to aid detection, it is not the actual "truth", as previous studies suggest that both reanalysis and GNSS data may contain inhomogeneities (Bock and Parracho, 2019; Zhang et al., 2019; Yuan et al., 2025). Moreover, in this study, we did not perform a separate changepoint search on ZTD, ZHD or Tm. Since PWV is derived from these parameters, any discontinuity in them can induce a corresponding offset in PWV, and we will further examine these in detail in future updates. **Additionally, to strengthen robustness, our future releases will cross-compare multiple detection methods (Van Malderen et al., 2020; Quarello et al., 2022; Nguyen et al., 2025) and adopt relative-homogeneity checks based on multiple pairwise comparisons (Caussinus and Mestre, 2004; Menne and Williams, 2009; Nguyen et al., 2024).**"*

19. Regarding alternative methods, Van Malderen et al., 2020, evaluated several of them in the similar context as the present study. Their benchmark study showed that PMTred, which is based on tests, is not performing well with the type of data used here (GNSS minus reanalysis differences). The best methods are indeed based on penalized maximum likelihood. One of them was first published under the name GNSSseg by Quarello et al., 2022, and recently updated and renamed PMLseg by Nguyen et al., 2025. This method uses penalized maximum likelihood and was specially developed for the segmentation of GNSS minus reanalysis differences. The authors may find it interesting as an alternative or simply for cross-checking their PMTred results.

**Response:** Many thanks for highlighting alternative offset-detection approaches. We are aware of the study by Van Malderen et al. (2020), which shows limited performance of PMTred for GNSS–reanalysis differences and indicates that penalised maximum-likelihood methods perform best. In line with this, we will cross-check our results using the GNSSseg/PMLseg (Quarello et al., 2022; Nguyen et al., 2025) and report the level of agreement (e.g., precision/recall of detected offsets and timing differences) in future releases.

As noted in our response above, we now add text acknowledging method sensitivity and outlining our plan to cross-compare multiple methods and adopt relative-homogeneity, multi-pair validation for corroboration. The current offset flags are provided alongside the PWV time series and we will update these files and document agreement metrics as the homogenisation effort progresses.

20. Figures 20 and 21: the interpolation of 2D fields over the ocean looks very unrealistic. It may be preferable to mask the oceans in these figures.

**Response:** Thank you for the constructive suggestion. Based on your (*mask ocean areas*), the two figures have been refined accordingly:

[Figure]

Figure 20. Rendered images of the 16-year climatological monthly mean PWV values for each month, derived from 590 GNSS sites located on the West Coast of the United States.

[Figure]

Figure 21. Rendered images of the annual mean PWV values for each year over the 16-year period 2006–2021, calculated from 590 GNSS sites located on the West Coast of the United States

21. Line 596: I was wondering if ENSO was not a more likely explanation for the interannual variability observed in these figures.

**Response:** Thank you for this insightful suggestion. The 2015–16 El Niño likely also contributed to the interannual variability. As noted in the manuscript, the "Blob" is an extratropical North Pacific marine heatwave and is distinct from (but interacting with) the ENSO, which is a tropical Pacific mode. Because these events co-occurred in 2015, and we cannot isolate their contributions to the phenomenon with the present analysis, we have moderated the wording (e.g., using "likely") and now state explicitly that both processes probably contributed to the observed peak. In addition, we have also added some supporting references here:

**Lines 646-649**

*"This peak likely represents the co-occurrences of the strong 2015/16 El Niño (L'Heureux et al., 2017) and the North Pacific marine heatwave known as "the Blob", a significant mass of relatively warm water in the northeast Pacific Ocean off the coast of the United States (Bond et al., 2015; Di Lorenzo and Mantua, 2016; Peterson et al., 2015). Together, these phenomena generated positive temperature anomalies exceeding 2.5 °C, with the warm ocean surface heating the overlying atmosphere."*

22. Section 7: Summary and Outlook: the text of this section seems to overstate a little the quality of the dataset given the mentioned limitations.

**Response:** We have moderated the wording in the revised version to avoid overstatement and have made the key limitations explicit:

**Lines 679-689**

*"This study has produced a global GNSS climate data record to help address data gaps in existing climate observing networks. Spanning up to a 22-year period from 2000 to 2021, the dataset includes hourly ZTD and PWV estimates from 5085 sites, providing broad spatiotemporal coverage and good overall accuracy globally. Advanced data reprocessing strategies, aligned with the IGS standards, were used to promote the consistency and accuracy of the generated atmospheric parameters, enhancing their suitability for climate applications. The quality of the dataset was evaluated via a rigorous quality assessment framework and cross-comparisons with various external references, including ERA5 reanalysis dataset, sounding profiles, and VLBI measurements. Generally good agreement across these datasets was demonstrated, with consistent water vapour estimates across diverse geographic and climate conditions. The dataset represents a critical step in GNSS climatology, offering valuable insights into the spatiotemporal variability of atmospheric water vapour. Further analyses of diurnal, monthly, seasonal, and annual variations in ZTD and PWV highlighted their importance in understanding climate variability, including responses to weather extremes and long-term climate trends.*

*......*

**Lines 711-715**

*Overall, the generated dataset represents a meaningful step toward fully harnessing the transformative potential of GNSS atmospheric monitoring techniques for advancing climate and atmospheric studies. By addressing critical challenges and leveraging cutting-edge methods, this dataset provides a reference for GNSS climatology, offering a foundation for future research and operational applications across this interdisciplinary field. These contributions may enhance our understanding of atmospheric dynamics,*

*supporting sustainable development and facilitating informed decision-making."*

> 23. While the dataset could benefit from further homogenization and the use of the latest GNSS processing software, this work represents a valuable contribution that will enrich discussions and collaborations within the IAG community (e.g. within the ICCC joint working group C.8 on Optimal processing and homogenization of GNSS-PW climate data records).

**Response:** We sincerely appreciate your encouraging and constructive comments. They are indeed helpful in further improving the quality of the manuscript and the generated dataset. We note the relevance to the ICCC joint working group C.8 and, in parallel, we are participating in a related working group (4.2.5) under the IAG frame. We are truly looking forward to future collaboration opportunities and hope to make our humble contributions to the community.

24. References

Bock, O. (2020) Standardization of ZTD screening and IWV conversion, in: Advanced GNSS Tropospheric Products for Monitoring SevereWeather Events and Climate: COST Action ES1206 Final Action Dissemination Report, edited by Jones, J., Guerova, G., Douša, J., Dick, G., de Haan, S., Pottiaux, E., Bock, O., Pacione, R., and van Malderen, R., chap. 5, pp. 314–324, Springer International Publishing, https://doi.org/10.1007/978-3-030-13901-8_5.

Bock, O., Bosser, P., and Mears, C. (2022) An improved vertical correction method for the inter-comparison and inter-validation of integrated water vapour measurements, Atmos. Meas. Tech., 15, 5643–5665, https://doi.org/10.5194/amt-15-5643-2022

Caussinus, H. & Mestre, O. (2004) Detection and correction of artificial shifts in climate series. Journal of the Royal Statistical Society Series C: Applied Statistics, 53(3), 405–425. Available from: https://doi.org/10.1111/j.1467-9876.2004.05155.x

Jones, J., Guerova, G., Douša, J., Dick, G., de Haan, S., Pottiaux, E., Bock, O., Pacione, R., van Malderen, R. (Eds.) ; Advanced GNSS Tropospheric Products for Monitoring Severe Weather Events and Climate: COST Action ES1206 Final Action Dissemination Report (2020), Springer International Publishing: Cham, Switzerland, 2020 ; https://doi.org:10.1007/978-3-030-13901-8.

Menne, M.J. & Williams, C.N. (2009) Homogenization of temperature series via pairwise comparisons. Journal of Climate, 22(7), 1700–1717. Available from: https://doi.org/10.1175/2008jcli2263.1.

Nguyen, K. N., Bock, O., & Lebarbier, E. (2024). A statistical method for the attribution of change-points in segmented Integrated Water Vapor difference time series. International Journal of Climatology, 44(6), 2069–2086. https://doi.org/10.1002/joc.8441

Nguyen, K. N., Bock, O., & Lebarbier, E. (2025), PMLseg: a R package for the segmentation of univariate time series based on Penalized Maximum Likelihood. https://github.com/khanhninhnguyen/PMLSeg/tree/main

Quarello, A., Bock, O., Lebarbier (2022) GNSSseg, a statistical method for the segmentation of daily GNSS IWV time series, Remote Sensing, 14, 3379, https://doi.org/10.3390/rs14143379

Van Malderen, R., E. Pottiaux, A. Klos, P. Domonkos, M. Elias, T. Ning, O. Bock, J. Guijarro, F. Alshawaf, M. Hoseini, A. Quarello, E. Lebarbier, B. Chimani, V. Tornatore, S. Zengin Kazancı, J. Bogusz (2020). Homogenizing GPS integrated water vapor time series: benchmarking break detection methods on synthetic datasets. Earth and Space Science, 7, e2020EA001121. https://doi.org/10.1029/2020EA001121

Venema, V., Mestre, O., Aguilar, E., Auer, I., Guijarro, J. A., Domonkos, P., et al. (2012). Benchmarking monthly homogenization algorithms. Climate of the Past, 8, 89–115. https://doi.org/10.5194/cp-8-89-2012.

**Response:** Many thanks for providing the detailed reference list. Most of these references are now being appropriately cited in the revised version.